# The Deployment Length of Solar Radiation Modification: An Interplay of Mitigation, Net-negative Emissions and Climate Uncertainty

Susanne Baur[1,2], Alexander Nauels[2,3], Zebedee Nicholls[3], Benjamin M. Sanderson[5], Carl-Friedrich Schleussner[2,4]

[1]CECI, Université de Toulouse, CERFACS, CNRS, Toulouse, 31100, France
[2]Climate Analytics, Berlin, 10969, Germany
[3]Australian-German Climate & Energy College, The University of Melbourne, Parkville, VIC 3010, Australia
[4]Geography Department and IRI THESys, Humboldt-Universität zu Berlin, Berlin, Germany
[5]Centre for International Climate and Environmental Research (CICERO), Oslo, Norway

*Correspondence to*: Susanne Baur (susanne.baur@cerfacs.fr)

**Abstract.** A growing body of literature investigates the effects of Solar Radiation Modification (SRM) on global and regional climates. Previous studies have focused on potentials and side-effects of SRM with little attention given to possible deployment timescales and the levels of Carbon Dioxide Removal required for a phase out. Here, we investigate the deployment timescales of SRM and how they are affected by different levels of mitigation, net-negative emissions (NNEs) and climate uncertainty. We generate a large dataset of 355 emission scenarios in which SRM is deployed to keep warming levels at 1.5°C global mean temperature. Probabilistic climate projections from this ensemble result in a large range of plausible future warming and cooling rates that lead to various SRM deployment timescales. In all pathways consistent with extrapolated current ambition, SRM deployment would exceed 100 years even under the most optimistic assumptions on climate response. As soon as the temperature threshold is exceeded, neither mitigation nor NNEs or climate sensitivity alone can guarantee short deployment timescales. Since the evolution of mitigation under SRM, the availability of carbon removal technologies and the effects of climate reversibility will be mostly unknown at its initialisation time, it is impossible to predict how 'temporary' SRM deployment would be. Any deployment of SRM therefore comes with the risk of multi-century legacies of deployment, implying multi-generational commitments of costs, risks and negative side effects of SRM and NNEs combined.

# 1 Introduction

Emission pathways that reflect the level of climate ambition of current Nationally Determined Contributions (NDCs) until the end of the century are estimated to lead to an average global warming of around 2.4°C (CAT, 2022). This is still a lot higher than the warming limit agreed in the Paris Agreement of 2015 that entails holding warming to well below 2°C and pursuing efforts of limiting warming to 1.5°C (UNFCCC, 2016). The growing concern around overshooting the Paris Agreement's long-term temperature target has led to a discussion of Solar Radiation Modification (SRM) which could in theory halt global

temperature increase very rapidly, but only as long as it is actively supported (Irvine et al., 2016; Keith, 2000). SRM techniques intend to artificially lower global mean surface air temperature (GSAT) by modifying the radiative energy budget of the Earth system. Proposed methods include Stratospheric Aerosol Injection (SAI), Cirrus Cloud Thinning (CCT) and Marine Cloud Brightening (MCB) (Lawrence et al., 2018). SRM methods generally operate on one of the key impacts of climate change, temperature increase, without addressing its cause, anthropogenic greenhouse gas (GHG) emissions, or other impacts e.g.

ocean acidification. Without explicit emission reductions as well as the removal of some of these climate forcers from the atmosphere in the long term, i.e. through Carbon Dioxide Removal (CDR) (Fuss et al., 2018), GHG emissions commit us to millennia of elevated temperature levels. Therefore, SRM deployment would only be temporary if combined with emission reductions and CDR.

    Achieving the Paris Agreement's target of 1.5°C relies on stringent mitigation with large near-term emission reductions as

shown in the 1.5°C-compatible pathways assessed by the Intergovernmental Panel on Climate Change (IPCC) (Rogelj et al., 2018; IPCC, 2021; IPCC, 2022). It has been discussed that, in the absence of this strong near-term mitigation, SRM could be a tool to avoid the impacts associated with overshooting 1.5°C until emission reductions and CDR are sufficiently scaled up that SRM is no longer needed to artificially lower GSAT (Belaia et al., 2021; Buck et al., 2020; MacMartin et al., 2018; Neuber & Ott, 2020; Allen et al., 2018). This 'buying-time'-approach, although criticised for relying on uncertain promises of SRM and CDR and increasing the risk of 'climate debt' (Asayama & Hulme, 2019), currently remains the dominant framing for any

SRM deployment (Neuber & Ott, 2020). Surprisingly little analysis, however, has been done on the timescales this type of SRM deployment could entail. Tilmes et al. (2016) analysed climate impacts of pathways whose temperature would peak at 3°C by the end of the 21[st] century and use CDR and SRM to limit temperature increase to 2.5°C and 2°C. Similarly, MacMartin et al. (2018) chose an experimental setup where mitigation, CDR and SRM are used to meet the 1.5°C-goal from a 'business-

as-usual' starting point. Both Tilmes et al. (2016) and MacMartin et al. (2018), did not discuss the length of SRM deployment and looked at selected illustrative pathways that cannot capture the many possible futures where a 'buying-time' approach of SRM could be embedded.

    In this study, we generate a large dataset of scenarios that use SRM to avoid overshooting the 1.5°C warming target. The underlying emission scenarios reflect current NDCs until 2030 and subsequently diverge to spread a wide range of conditions

by the end of the century. We employ this large scenario set to explore a large variety of futures of SRM interlinkages with

mitigation ambition and different magnitudes to which CDR could be scaled up. There is large uncertainty regarding the evolution of emissions under SRM with some studies arguing that SRM could be deployed for 'peak-shaving' under already ambitious mitigation scenarios (Coninck et al., 2018) or does not negatively affect the public's willingness to engage in mitigation behaviours (Andrews et al., 2022; Austin & Converse, 2021; Fairbrother, 2016; Kahan et al., 2015; Merk et al., 2016), while others fear it could undermine mitigation ambition even further (Baatz, 2016; Corner & Pidgeon, 2014; Pierrehumbert, 2019; Raimi et al., 2019) and present a 'moral hazard' risk (Belaia et al., 2021; Bellamy et al., 2016; Burns et al., 2016; Keith, 2000; McLaren, 2016; Merk et al., 2016; Moreno-Cruz, 2015; Wibeck et al., 2015). Here, we do not investigate to what extent SRM will or will not change mitigation ambition and instead highlight what various emission reduction assumptions could mean in terms of SRM deployment length. Thanks to the large dataset underlying this study, we can analyse several other factors that influence the length of SRM deployment such as the amount of annual net-negative emissions (NNEs) realised through large-scale CDR and climate system uncertainty.

There is a large body of literature dedicated to CDR and its application in mitigation pathways (Fuss et al., 2014, 2018; Johansson et al., 2020; Rogelj et al., 2019; Schleussner et al., 2016). Large uncertainties regarding overshoot pathways and CDR remain, especially since the technology and the resulting temperature declines are unproven at scale (Pathak et al., 2022). Uncertainties are related to the response of the climate system to negative emissions, the state of the climate system post overshoot in general, environmental and economic side-effects of large-scale deployment, and the level to which CDR can be scaled up (Fuss et al., 2018; Matthews et al., 2020; Rogelj et al., 2019; Schleussner et al., 2016; Zickfeld et al., 2016). In this study, we do not account for feasibility constraints or potential side-effects of CDR deployment, but rather explicitly assess the sensitivity of our results to any such constraints being in place. As part of our sensitivity assessment, we consider a wide range for maximum annual NNEs reaching up to 40 $GtCO_2$/yr (the highest potential the IPCC 6th Assessment Report (AR6) assigns to any carbon removal technology that does not interfere with ocean chemistry (IPCC, 2022)) as well as the uncertainty surrounding the temperature decline in response to net-negative emissions.

It is important to examine SRM deployment length in the context of climate uncertainty. In contrast to mitigation and CDR, climate uncertainty is beyond human control and, as this study demonstrates, would result in greatly differing SRM deployment outcomes for the same levels of emissions. We address climate uncertainty in two ways. First, by considering a large range of plausible climate simulators. And secondly, by calculating two climate metrics that relate emissions to temperature change and affect the duration of the overshoot and therefore SRM deployment length.

While many different methods for CDR (Fuss et al., 2018) and SRM (Boucher et al., 2013; Lawrence et al., 2018) exist that come with different specificities, this study does not differentiate between these specific technological approaches as our results are independent of the boundary conditions for individual SRM and CDR techniques. We acknowledge that forecasting technology this far into the future is highly speculative and this analysis is by no means intended to be a realistic representation of SRM and CDR pathways. Therefore, this paper does not address issues of feasibility or environmental side effects of SRM

or CDR, of which there are many (Lee et al., 2021; Canadell et al., 2021; Douville et al., 2021). Neither does it propose potential implementation strategies and designs or poses questions relating to economic, political or ethical concerns. With this contribution, we aim to provide a conceptual framework for exploring SRM deployment length in the context of scenarios that use the technology as a temporary (albeit potentially multi-century) measure.

## 2 Methods

### 2.1 Emissions data and pathway extension

As underlying data, we use all scenarios from the IPCCs' 6th Assessment Reports' database that are in line with the 2030 NDCs (Riahi et al., 2022; Byers et al., 2022) and have decreasing or stagnating emissions in the last five years of the 21st century. The policy categories we consider when identifying the scenarios that pass through NDC emissions are P3b, P1c and P0_3b (Riahi et al., 2022). This amounts to a total of 355 scenarios covering the years 2015 to 2100 that originate in a similar climatic state in 2030 but diverge afterwards to cover a large variety of emission levels in 2100. We employ this broad range of pathways to analyse SRM under many different developments of mitigation ambition under SRM, such as large increases and decreases, as well as under the scrutiny of climate change uncertainty by running a probabilistic ensemble of 600 members (see 2.2).

To be able to estimate potential SRM timescales, we extend all pathways until 2500. In order to explore a wide range of possible future development, we randomly sample three parameters for the extension of each scenario: the change in rate of decarbonisation after 2100 (0-3% increase), the maximum net-negative fossil-$CO_2$ emissions (log-normal distribution from 0-40 $GtCO_2$/yr) realised through large-scale CDR (Figure 1c) and the floor for $CO_2$ emissions from agriculture, forestry and other land use (AFOLU-$CO_2$) (normal distribution -1-+1 $GtCO_2$/yr) (Figure 1f). We choose a log-normal distribution of 0 to 40 $GtCO_2$/yr with positive skew for net-fossil-CDR. The range depicts the maximum mitigation potential that the IPCC AR6 assigns to industrial carbon removal technologies (Direct Air Carbon Capture and Storage (DACCS)) while the distribution with positive skew represents the tendency of limits to be at the lower rather than higher end of the chosen spectrum. The mean of the distribution is set to be 15 $GtCO_2$/yr, representing the rounded up median potential experts assign to DACCS in 2100 (Grant et al., 2021). The storage capacity of technologically captured $CO_2$ is uncertain but likely in the range of 8000 to 55000 $GtCO_2$ (Dooley, 2013; Kearns et al., 2017). This capacity potential does not impact our results because even the lower end is sufficient for the majority of our pathways (Figure 4b). We therefore choose no constraint on maximum negative cumulative fossil-$CO_2$ emissions. In contrast, nature-based CDR solutions represented by AFOLU negative emissions are seriously limited in the total amount they can remove. Current literature suggests that AFOLU-CDR is a solution of the 21st century and that saturation could be hit soon after (Fuss et al., 2018). Therefore, we remove the need to make decisions about the role of AFOLU-CDR (which would be a relatively small part of the picture over the timescales we are considering) by setting the AFOLU long-term emissions level to be close but not exactly equal to 0 (-1 to 1 $GtCO_2$/yr in line with the Global Carbon

Budget 2022 (Friedlingstein et al., 2022)). In the analysis of our results, we sum AFOLU and fossil net-negative emissions to one variable which we call Net-Negative Emissions (NNEs).

The modified decarbonisation rate of the last ten years of the 21st century is linearly extrapolated until meeting the maximum net-negative fossil-$CO_2$ emissions for the respective pathway (Figure 1b,e). In the rare cases where scenarios have increasing AFOLU- or fossil-$CO_2$ emissions in the last 10 years (we only allow scenarios with stagnating or decreasing *combined* $CO_2$ emissions in the last few years, meaning fossil-CDR or AFOLU-CDR individually can still have increasing emissions or decreasing negative emissions in the last decade) are assigned decreasing emissions after 2100 with the rate of increasing emissions they had before plus the change in rate that is randomly prescribed (0-3%) (see uppermost lines in Figure 1b,e). We assume that saturation for nature-based carbon removal is hit in 2150 and all pathways linearly move towards their randomly chosen floor level after 2100 with a rate that allows reaching the assigned floor by 2150 (Figure 1e). For simplicity, we hold non-$CO_2$ emissions constant at 2100 emission levels until 2500, which gives some residual warming signal but is relatively small compared to the $CO_2$ contribution.

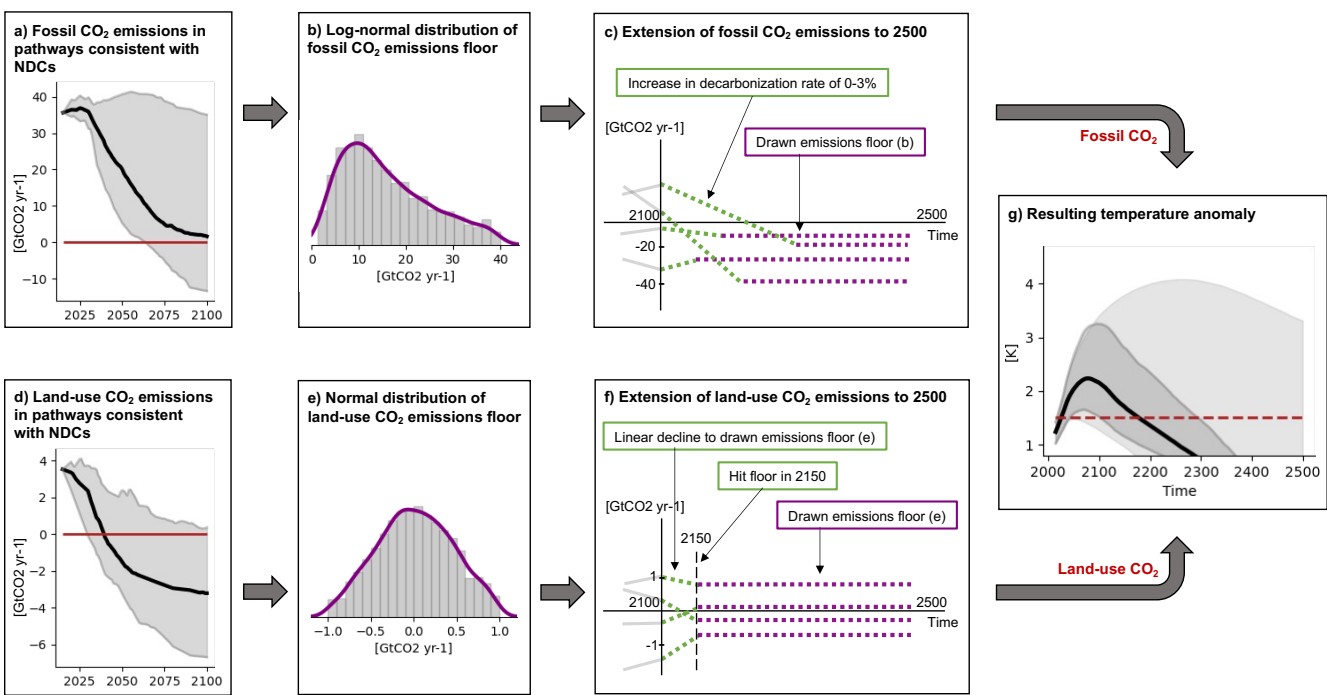

*Figure 1: Extension of emission pathways consistent with NDCs from the AR6 WG3 database to 2500. A distribution of extension options is used to sample the range of possible outcomes consistent with current literature. For simplicity, in all cases, non-$CO_2$ emissions are kept constant after 2100. a) Fossil $CO_2$ emissions from selected IPCC AR6 WG3 database scenarios (displayed 5th-95th percentile). b) Distribution of minimum fossil $CO_2$ emissions used in this study. c) Illustration of extension algorithm used for fossil $CO_2$ emissions - each pathway is extended at a fixed rate of decline until it hits some*

*prescribed value, after which emissions are held constant (see 2.1 for full description). d) Land-use $CO_2$ emissions from selected IPCC AR6 WG3 database scenarios (displayed 5th-95th percentile). e) Distribution of steady state land-use $CO_2$ emissions used in this study. f) Illustration of extension algorithm used for land-use $CO_2$ emissions – each pathway is extended such that it reaches a specific value by 2150, after which emissions are held constant (see 2.1 for full description). g) resulting temperature trajectories, light gray covering the 5th-95th percentiles over all scenarios with all 600 ensemble members, dark gray the 5th-95th percentile of an exemplary scenario (MESSAGEix-GLOBIOM_1.1 EN_INDCi2030_1800f_NDCp) with the black line being the median of it. We span a wide range of trajectories, from always remaining below 1.5°C to never coming back below 1.5°C after crossing 1.5°C in the early 21st century.*

## 2.2 MAGICC setup and SRM-pathway construction

All global climate model simulations are conducted with the climate emulator MAGICC7.5.3 (Meinshausen et al, 2020). This emulator is commonly used in several leading Integrated Assessment Models and consecutively in the IPCC assessments, including the most recent 6th Assessment Cycle. The model includes a simplified terrestrial and ocean carbon cycle (Meinshausen et al., 2009, 2011, 2020). We apply a probabilistic setup with an ensemble of 600 runs derived by a Markov-Chain Monte-Carlo approach and display all ensemble members except if indicated otherwise. The range of the ensemble members depicts the Equilibrium Climate Sensitivity uncertainty range of the IPCC 6th Assessment Report (Forster et al., 2021) and the C4MIP carbon cycle ranges (Forster et al., 2021) and as a consequence offers good coverage of climate system and model uncertainty. The ensemble members span a Transient Climate Response to Cumulative Carbon Emissions (TCRE) range of 0.87 to 3.47 [K / 1000 PgC] (17-83% range 1.37-2.19).

The purpose of SRM in our scenario setup is to cool the temperature overshoot pathways down to global average warming of 1.5°C starting in 2030. Because SRM is implemented in the model by modifying the Effective Radiative Forcing (ERF), it is necessary to determine what forcing pathway is equivalent to following a 1.5°C-compliant trajectory for each member of the ensemble (Figure 2). This 1.5°C-trajectory represents the 'SRM-pathway'. Due to computational efficiency and close resemblance to WG3 NDC-pathways, we choose an SSP2-4.5 pathway as a starting point for this SRM-pathway construction (Figure 2a). This reduces the required computing time by a factor of 10 while still retaining sufficient accuracy to make useful quantifications of required SRM deployment times. Using 2030 NDC emission levels as the starting point for our analysis, we also assume that radiative forcing can be modified no sooner than 2030. Depending on the ensemble member this leads to either a smooth approach to 1.5°C where an overshoot is avoided in the SRM-pathway (Figure 2b) or to an overshoot that is subsequently brought down to 1.5°C (Figure 2c). Whether the SRM-pathway for a given ensemble member overshoots is determined by its SSP2-4.5 2035 warming level had it followed its 10-year gradient from 2030 to 2040 for five more years after 2030. If this 2035 warming level is higher than 1.5°C, the pathway will overshoot to the respective 2035 level and subsequently descend to 1.5°C in a sigmoidal pattern to reach 1.5°C in 2050.

Since each of the 355 scenarios is prescribed different assumptions (see 2.1) and each ensemble member in MAGICC's probabilistic distribution has different physics, we must calculate the SRM on a scenario-ensemble member basis where the SRM required is the difference between the 1.5°C-pathway and the extended NDC-scenario without SRM (NDC-extension described in 2.1). The start date of SRM is determined by the date where the extended NDC-scenario exceeds the designed SRM-pathway for the respective ensemble member. SRM termination is assumed to happen once 1.5°C global mean warming is reattained in the NDC-scenario.

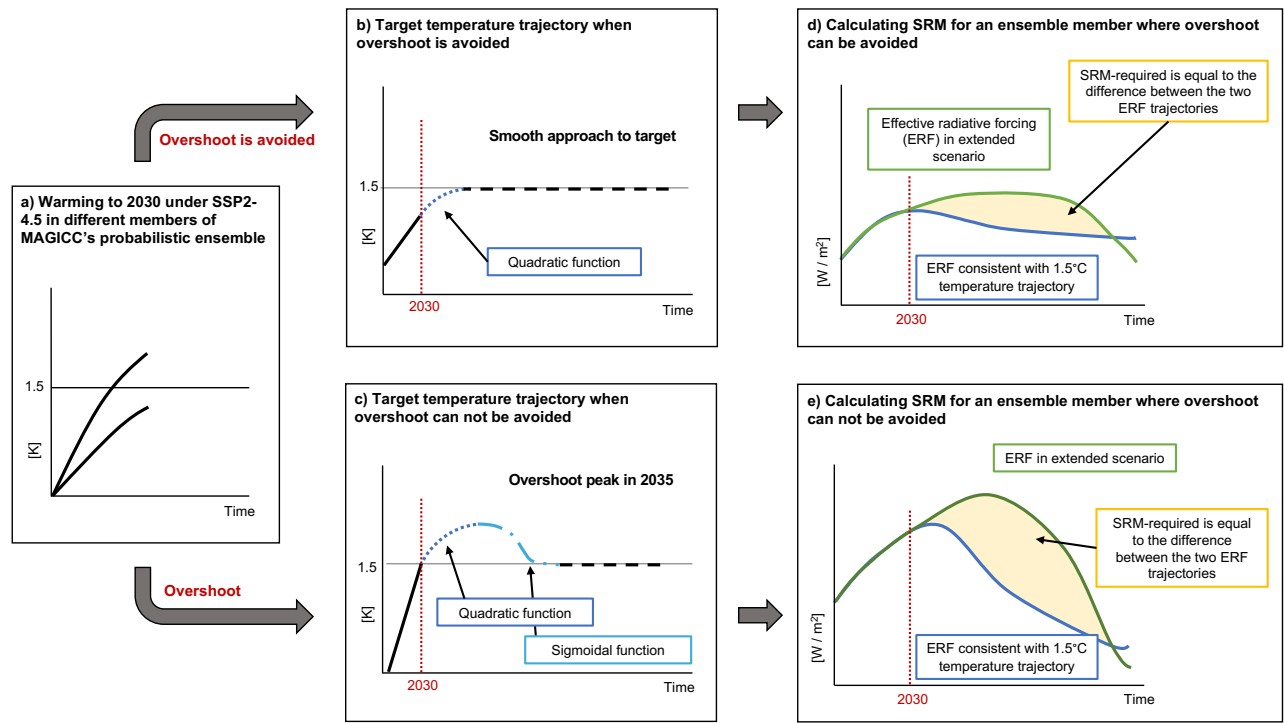

*Figure 2: Calculating required SRM. a) Calculating warming to 2035 using an NDC-like pathway (in this case SSP2-4.5). b) Determining a 1.5°C temperature trajectory for ensemble members that have not already overshot 1.5°C by 2035. c) Determining a 1.5°C temperature trajectory for ensemble members that have already overshot 1.5°C by 2035. d), e) Calculating required solar radiation manipulation (SRM) for each scenario-ensemble member combination, whether it overshoots 1.5°C (e) or remains below 1.5°C at all times (d).*

To address climate uncertainty and depict the whole range of possible outcomes, we use all members of our ensemble instead of only focusing on medians. Additionally, we calculate the uncertainty surrounding the rise in temperature for a specific amount of $CO_2$ emissions and non-$CO_2$ forcing, called effective Transient Response to Cumulative $CO_2$ Emissions (eTCRE,

subsequently called eTCRE-up) (Gregory et al., 2009; Matthews et al., 2009, 2020; Vakilifard et al., 2022). We also consider uncertainty in the temperature change as a result of reducing the concentrations of $CO_2$ and associated non-$CO_2$ gases in the atmosphere (subsequently called eTCRE-down). Due to uncertain Earth system feedbacks such as a lagged ocean response, it is possible that eTCRE-up and eTCRE-down are not identical (Matthews et al., 2020; Zickfeld et al., 2016) and this asymmetry is reflected in the MAGICC ensemble. eTCRE-up informs whether temperature targets are exceeded for a specific amount of cumulative emissions and therefore whether SRM is deployed in our scenarios or not, as well as which peak warming is hit when. eTCRE-down informs us on how effective NNEs are at cooling, meaning how many cumulative NNEs we need and how long it takes to return to our target temperature increase of 1.5°C.

We define eTCRE-up and eTCRE-down as:

(1) $eTCRE_{up} = \dfrac{\Delta T_{2030 \rightarrow peak\_warming}}{\Sigma_{2030}^{peak\_warming} CO_2\ emissions}\ [K\ 1000PgC^{-1}]$

(2) $eTCRE_{down} = \dfrac{\Delta T_{1.5°C \rightarrow peak\_warming}}{\Sigma_{peak\_warming}^{year\_of\_return\_to\_1.5°C} CO_2\ emissions}\ [K\ 1000PgC^{-1}]$

While the change in temperature is related to both $CO_2$ and non-$CO_2$ gases, the cumulative emissions are only related to $CO_2$. Due to a few extreme outliers both metrics are constrained to their 1-99th percentile.

## 3 Results

The 355 NDC-scenarios including extensions and probabilistic MAGICC7 simulations lead to 213 000 different realisations with a large range of warming outcomes throughout the centuries (Figure 1g). While few realisations peak at or below 1.5°C, the vast majority of the baseline simulations without SRM overshoot the temperature target temporarily. Others do not return back to 1.5°C before 2500 at all.  The 5-95th percentile range of peak-warming in our dataset ranges from 1.60°C to 3.91°C with the median at 2.14°C. Therefore, even though all realisations originate in 2030-NDC levels, due to the range of possible developments of mitigation and NNEs under SRM, the resulting SRM deployment length ranges from 0 to >470 years (Figure 3), with 5% of realisations not requiring SRM to limit warming to 1.5°C and 15% having deployment times that exceed 470 years. In the following analysis we only include realisations that fall into the 1-470 year range.

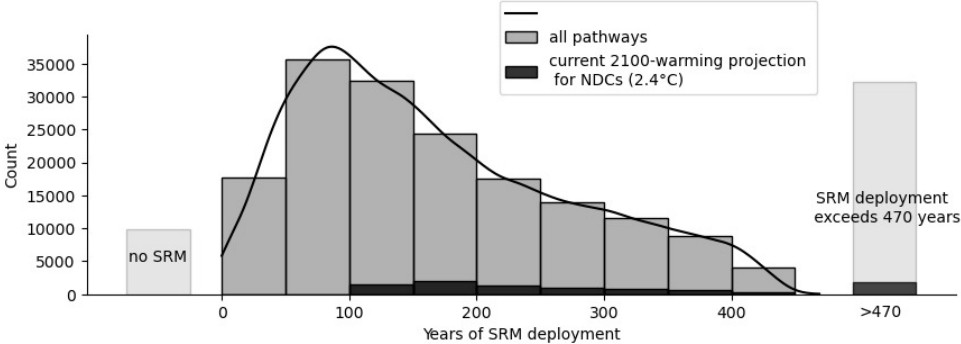

*Figure 3: SRM deployment length for all scenarios and all ensemble members. 1 bar spans a range of 50 years. Marked in black are pathways consistent with current 2100-warming projections for NDCs (2.4°C) (CAT, 2022).*

Figure 4 breaks down the three indicators mitigation, negative emissions and climate uncertainty and their interlinkages with SRM deployment length. The relationship between the emission pathway, i.e. the cumulative emissions from 2030 until net-zero, and SRM deployment length is weak (Figure 4a). The triangular shape of the data points towards a tendency for very high cumulative emissions (>3000 GtCO$_2$ at time of net-zero) to lead to longer SRM timescales, whereas scenarios with lower cumulative emissions are spread across the whole range of SRM deployment times with a skew towards the shorter end. Realisations above 1500 GtCO$_2$ (apart from a few exceptions) require at least a few decades of deployment to keep warming at 1.5°C, no matter the climate uncertainty. The plot looks stratified because the cumulative emissions value is member-independent and therefore stays equal across all ensemble-members of each scenario.

The way our scenarios are set up, the choice of maximum potential NNEs is random and therefore independent of the amount of cumulative emissions. However, *average* annual NNEs are not entirely arbitrarily spread: shorter deployment timescales are constrained to lower annual NNEs because of the limited time these scenarios have to scale up their net-negative emissions. Similarly, high emission scenarios take longer to reach high net-negative emissions. This is demonstrated by scenarios with cumulative emissions above 3500 GtCO$_2$ that almost all reach annual NNEs above 15 GtCO$_2$/yr by 2500 (see Figure S01 in S.I.) but most never reach this level in average terms because of the time required to scale up.

Focussing on the interlinkages with negative emissions, Figure 4b shows cumulative emissions between net-zero and reattaining 1.5°C against SRM deployment. The relationship is much clearer and the triangle shape of Figure 4b considerably more defined; low cumulative negative emissions are represented across the whole range of SRM deployment length but high cumulative negative emissions are constrained to long SRM timeframes. For example, cumulative negative emission requirements above -2000 GtCO$_2$ imply more than 100 years of SRM deployment. Similarly, -6000 GtCO$_2$ lead to more than 200 years of SRM deployment and -10000 GtCO$_2$ to more than 300 years. Part of this effect is the simple fact that you only get very large amounts of cumulative negative emissions if you're deploying SRM for long time periods.

Where in Figure 4a the annual average NNEs are partly random, a clear pattern becomes visible in 4b. The higher the annual average NNEs the shorter the SRM timescale for the same amount of cumulative emissions. For higher total negative emission requirements low NNEs are not sufficient in limiting the deployment to 470 years and are thus not shown. Very small amounts of NNEs (< 1 $GtCO_2$/yr) are constrained to pathways with no or very small amounts of required negative emissions to get down to 1.5°C of warming.

Regarding climate uncertainty, the calculated 5-95th percentile range of eTCRE-up is 1.0 - 6.0 [K/1000PgC], eTCRE-down is -6.9 - 12.4 [K/1000PgC]. Figure 4c sets the eTCRE-ratio (defined as eTCRE-up / eTCRE-down) against the SRM timeframes. The spread in uncertainty increases with SRM deployment length. Realisations with an absolute eTCRE-ratio of >1 have a lower sensitivity to negative emissions than positive ones, causing them to deploy SRM for longer periods even though net-zero is hit earlier than for other scenarios (see Figure S02 S.I.). Negative eTCRE-ratios (dark purple data points) are the result of negative eTCRE-down values, all of which arise in realisations that have positive cumulative emissions in the cooling phase. This could be due to a high negative Zero Emissions Commitment (ZEC) (Jenkins et al., 2022; C. D. Jones et al., 2019; MacDougall et al., 2020) or non-CO2 effects that allow temperatures to drop despite cumulative positive $CO_2$ emissions.

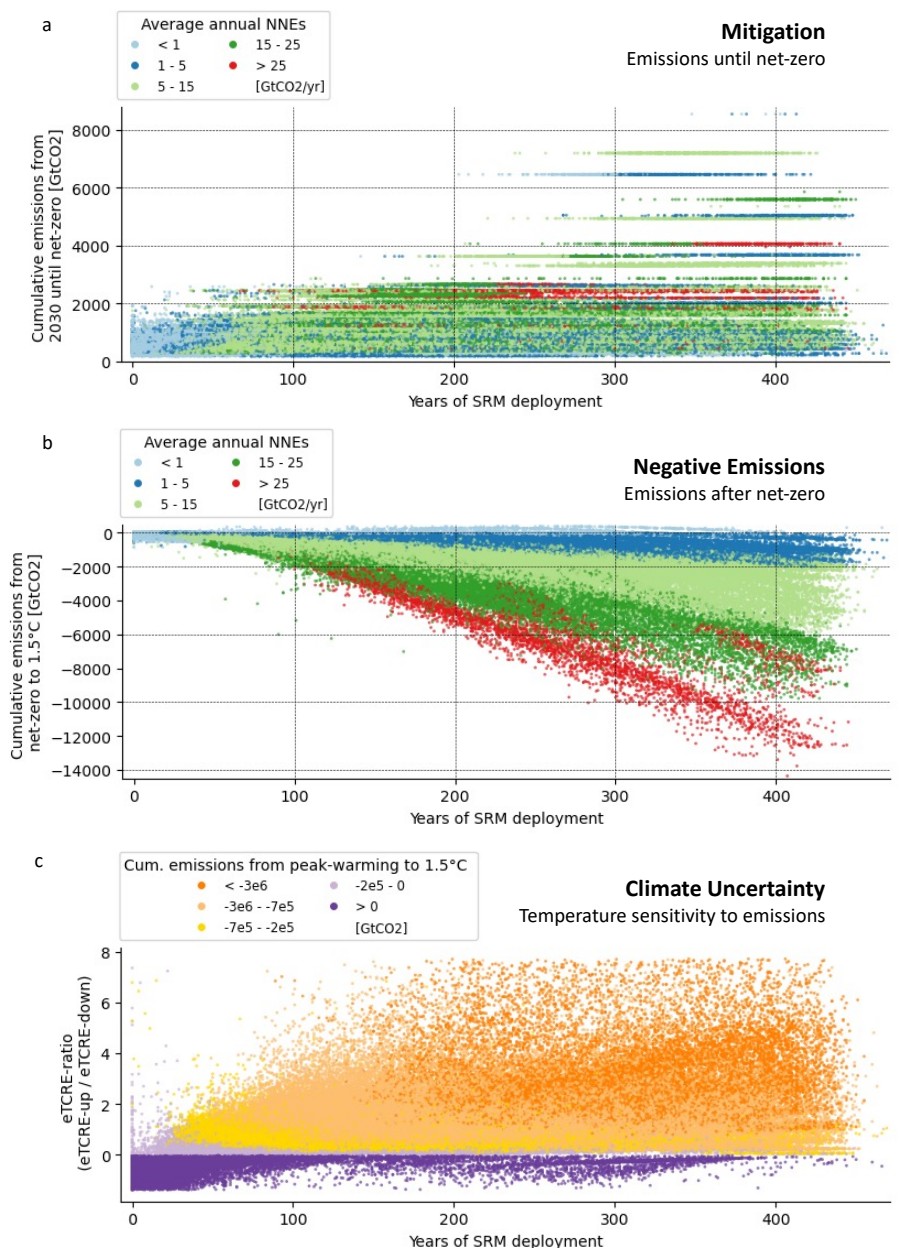

260

*Figure 4: Interdependencies of mitigation, negative emissions and climate uncertainty with SRM deployment length a) Relationship between cumulative $CO_2$ emissions from 2030 until net-zero $CO_2$ and SRM deployment length. Colour-coding according to annual average NNEs in GtCO2/yr. b) Relationship between cumulative $CO_2$ emissions from net-zero $CO_2$ until reattaining 1.5°C and SRM deployment length. Colour-coding according to annual average NNEs in GtCO2/yr. c) Relationship*

*between eTCRE-ratio and SRM deployment length. Colour-coding according to cumulative $CO_2$ emissions from time of peak-warming until reattaining 1.5°C. Plot shows data points that fall in the 1-99th percentile range.*

## 4 Discussion

In current literature, SRM is often framed in the context of a stopgap measure (Asayama & Hulme, 2019; Buck et al., 2020; Neuber & Ott, 2020). Our study focuses on the question of what range of timeframes would be consistent with an intended temporary SRM deployment due to uncertainty in mitigation ambition, negative emissions and climate uncertainty.

We show that the range of possible deployment timescales is vast even for pathways that have similar conditions at the start of SRM deployment, in our case in 2030 (Figure 3). This is due to the uncertain evolution of mitigation ambition and annual NNEs under SRM and the effects of climate uncertainty. We find that neither of these three indicators (mitigation, net-negative emissions or climate uncertainty) alone can determine SRM deployment length. Mitigation and negative emissions represent a bounding condition on SRM deployment length rather than a linear correlation, to some degree due to the climate sensitivity of increasing and decreasing emissions (Figure 4c). This means that positive emissions after SRM initialisation on the medium to lower end could imply both short and long term SRM deployment (Figure 4a). Similarly, low cumulative negative emission requirements are no guarantee for short SRM timeframes (Figure 4b). However, the faster and higher annual NNEs are scaled up the shorter the time-wise commitment to SRM (Figure 4b, 5a). Our data suggest that for the range of emission pathways, NNEs and in particular the limited analysis length of 470 years in our experiments, it might be easier to determine a lower limit of deployment length than an upper limit.

The relationship between the eTCRE-ratio and SRM deployment length versus cumulative emissions that results from our data is complex and requires further study (Figure 4c, 5b). Several aspects play into the variables used to calculate the two climate uncertainty metrics. For example, the temperature change in the eTCRE calculation is related to $CO_2$ as well as non-$CO_2$ gases while the cumulative emissions are only a function of $CO_2$. Additionally, the experimental setup of this study is such that $CO_2$ and non-$CO_2$ emissions are not linearly related: Non-$CO_2$ emissions vary between 2030 and 2100 and then stay constant at the 2100 level while $CO_2$ emissions fall consistently after 2100.

Our calculated eTCRE-up is higher than the pure range of TCRE that the MAGICC7 ensemble members have, i.e. the TCRE range in the AR6 of the IPCC (0.87-3.47). This is expected and has been observed before because of the impact of non-$CO_2$ climate forcers (Matthews et al., 2020, 2021). Assumptions regarding the constant ratio of cumulative emissions of $CO_2$ to temperature (TCRE) are suggested to go up to at least 3000 PgC of positive cumulative $CO_2$ emissions (Leduc et al., 2015; Tokarska et al., 2016). Our values exceed this limit. However, this might not be directly transferable to our eTCRE-up metric especially since these values are expectedly higher than TCRE (Matthews et al., 2020). Studies with intermediate complexity climate models have found a higher TCRE-up than TCRE-down, suggesting a greater impact of positive $CO_2$ emissions on temperatures than negative emissions (Zickfeld et al., 2016). This can be explained by a lagged ocean response that leads to continued warming after the start of carbon removal (Zickfeld et al., 2016). However, ocean effects might differ in cases where

SRM is deployed and complex earth system models show a range of responses after emissions are halted (Jenkins et al., 2022; MacDougall et al., 2020), let alone negative for a sustained period.

The inconclusive relationship of the eTCRE-ratio and negative emissions against SRM deployment length (Figure 5) emphasises the fact that while ending SRM hinges fundamentally on the decline of temperatures (when SRM is used for the purposes described in this paper), climate reversibility is currently marked by large uncertainties.

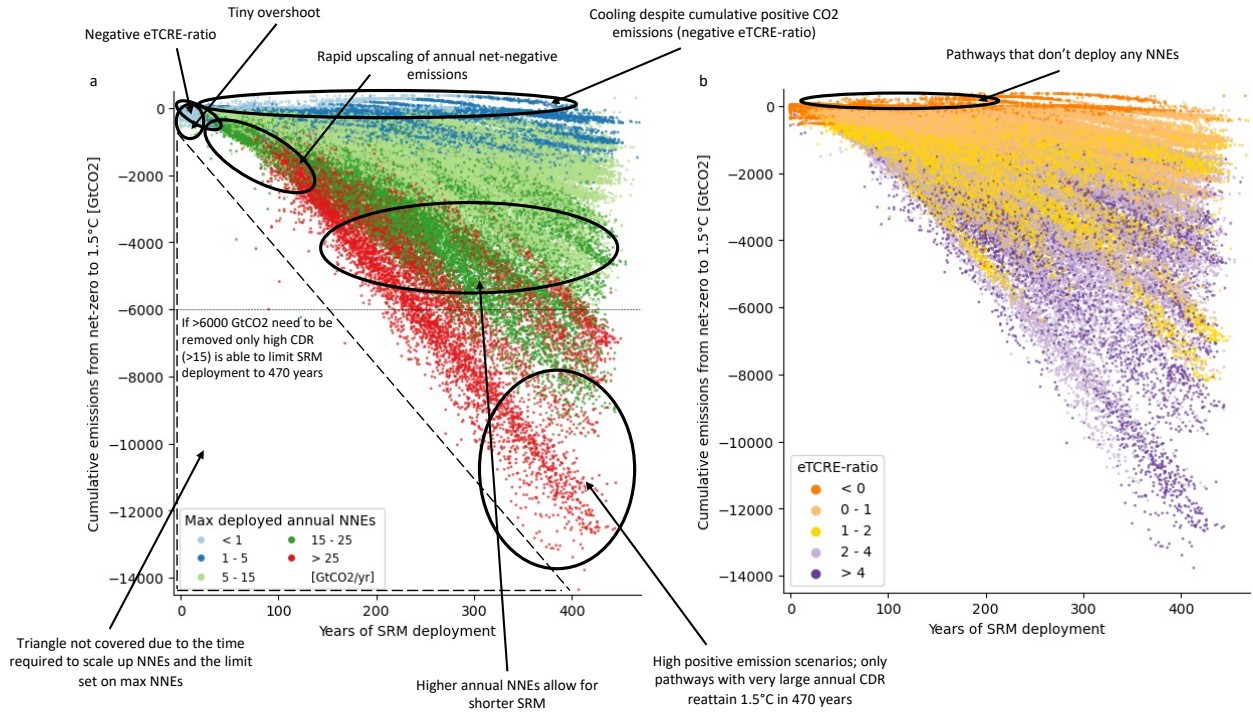

Figure 5: *Relationship between cumulative $CO_2$ emissions from net-zero $CO_2$ until reattaining 1.5°C and SRM deployment*
*length. a) Colour-coded by the maximum deployed annual NNEs. b) Colour-coded by the eTCRE-ratio. Clean figure without*
*descriptions in S.I (Figure S03).*

The shape of the triangle in Figure 4a,b and Figure 5a,b is partly due to the effects of climate uncertainty and partly due to the experimental setup with a limit of yearly NNEs at 40 $GtCO_2$/yr. However, the area of the plot with short SRM deployment
under around a century would not be covered even for higher amounts of annual NNEs because the scaling up requires time. To contextualise, a yearly removal of 40 $GtCO_2$ equates to today's yearly fossil $CO_2$ emissions (Friedlingstein et al., 2022) and would entail a massive industrial effort by itself, which would take decades to build up. So even though very high NNEs could shorten SRM timescales, it likely would not be available for short SRM deployments. In those cases, the best bet is to

require a small amount of total net-negative emissions to return back down to 1.5°C. This would imply low cumulative positive
emissions and a high eTCRE-down.

The upper limit of NNEs goes to the heart of concerns around CDR: the magnitude of yearly removal to which it can be scaled up. Because it is impossible to predict technological development this far into the future we reflect all plausible options by assuming NNEs that range from barely being able to compensate residual positive emissions, to being consistent with current sustainable levels, to much larger amounts that future technological development could enable but far exceed many estimates
in current literature (Fuss et al., 2018; Grant et al., 2021; Coninck et al., 2018; Pathak et al., 2022). It needs to be highlighted that this study looks at net-negative emissions and not CDR. This means that our estimates of net-negative emissions are a lower bound on the amount of CDR needed. There is a broad discussion in current literature on negative side effects and sustainability concerns with CDR in mitigation scenarios (Brack & King, 2020; Fuss et al., 2018; Smith et al., 2015) as well as the question of equity in the allocation of CDR burden (Fyson et al., 2020). Major concerns are the considerable land, water,
energy and financial requirements and constraints in long-term storage of removed $CO_2$ that increase for higher yearly removal rates. These recognized environmental and social concerns of CDR are at least as applicable for pathways where CDR exceeds current sustainable ranges and needs to be sustained for decades up to centuries. This paper quantifies the importance of decades-long large-scale CDR for 'peak-shaving' SRM scenarios, which has previously been pointed out by Asayama & Hulme (2019) and others. In addition, we find that initialisation and commitment to SRM happen under the promise that CDR
can be scaled up high enough to end SRM again. Belaia et al., 2021 even demonstrate that a cost-optimal portfolio to meet 2°C GSAT increase would consist of SRM and mitigation first and later CDR deployment. With CDR being unproven at scale and peak-and-decline scenarios marked by large uncertainties, this is a risky promise. Our scenarios show what happens if the scalability of CDR is not as high as assumed or TCRE-down is low: SRM deployment will have to be extended for a much longer period of time (Figure 4b).

In scenarios where the level of warming in 2100 follows the level of mitigation ambition reflected in 2030 NDC commitments until the end of the century (CAT, 2022), no pathway deploys SRM for a shorter period than 100 years, while most require 150-300 years in order to return to 1.5°C GSAT increase (black bars in Figure 3; 25-75% percentile range 159-294 years, excluding all pathways that exceed 470 years in the statistic). Even for scenarios with a somewhat smaller overshoot above 1.5°C potential deployment length remains considerable. Half of all scenario realisations that would peak around 1.8 °C require
SRM deployment for 96-195 years to stay at 1.5°C. Therefore, 'short' peak-shaving SRM deployment perhaps implies a length of the order of 100-200 years rather than 10-50. These timeframes are much longer than the hotly debated CDR deployment lengths in GHG emission reduction pathways for limiting warming to 1.5°C in 2100 after a high or no/limited overshoot assessed in the IPCC SR1.5 (Rogelj et al., 2018) or IPCC AR6 WG3 (Riahi et al, 2022).

MacMartin et al. (2018) created an exemplary scenario where emissions would lead to a peak warming of 2.7°C without SRM
and deploy 15 Gt CDR per year. Their scenario requires around 235 years of SRM deployment to limit warming to 1.5°C. Constraining our dataset to their benchmark data results in an SRM deployment of 99-404 years with a mean at 221 - a range

that encompasses their result with a mean that is not far off from their estimate. Tilmes et al. (2016) constructed a scenario that peaks at 3°C and uses SRM to cool down to 2.0°C with 18.5 GtCO$_2$ CDR per year. Simulated with the CESM Earth System Model they indicate a deployment time of 160 years to compensate the overshoot over 2°C. While these results are not fully comparable, compensating 1°C of overshoot with NNEs of 18.5 GtCO$_2$ per year in our scenarios results in an average deployment length of 204 years and a full range of 98-393 years. Our results thus compare well with previous explorations of this question documented in the literature, yet provide a much more holistic perspective on the uncertainties involved. As we demonstrate, these are just few of the many possible future outcomes of SRM deployment and including climate uncertainty shows where the limits of our control lie.

The challenge in defining SRM deployment length at its initialisation poses a clear risk as there are several issues that might arise with a multi-century SRM deployment: One of the arguably biggest difficulties of SRM deployment is international cooperation and coordination (Möller, 2020; Reynolds, 2019; Shepherd et al., 2009). Maintaining international cooperation will be even more difficult if deployment needs to be sustained over many decades as priorities and interests of countries and their leaders might change. Furthermore, long-lasting SRM deployment would require substantial financial resources; due to the difficulty in predicting SRM deployment length as pointed out in this study, costs could end up being much higher than originally anticipated if deployment ends up being longer than planned. This needs to be put into context with the overall cost of responding to climate change and avoided costs of climate change damages through SRM (Arino et al., 2016; Belaia et al., 2021). Moreover, a key risk of SRM is the so-called 'termination shock', a rapid warming response to a sudden stop of SRM deployment (Parker & Irvine, 2018). The longer the deployment, the longer the risk for such an occurrence. And lastly, long-term deployment would bind future generations to SRM which raises substantial ethical and moral questions (Flegal et al., 2019; Goeschl et al., 2013; Svoboda et al., 2011). The idea of imposing a technology on individuals who have not yet been born and do not have a say in the matter can be considered to be a violation of their rights and autonomy. This ethical risk needs be considered in conjunction with the additional climate change-related risks from ongoing warming also imposed on future generations who did not contribute to the problem to begin with.

In addition to all the risks, costs and potential side-effects that come with SRM, due to the dependency, future generations would be burdened by large-scale deployment of CDR which might compete with efforts to secure their own requirements by increasing risks of biodiversity loss and food and water scarcity (Dooley & Kartha, 2018; Shue, 2017; Asayama & Hulme, 2019; Svoboda et al., 2011). With this study we add quantitative data to the literature calling for precautionary and ethical approaches to technology development. The identified risks need to be weighed against the risk of not deploying SRM. Our research highlights the substantial dependencies of SRM and CDR deployment which imply side-effects, risks, costs and uncertainties of both SRM and CDR.

In this analysis, we use SRM for the prevention of overshooting the 1.5°C-target and the deployment lengths we indicate only hold under the stated conditions of the analysis. Possible avenues for prior phase-outs have been discussed in Keith &
MacMartin (2015), MacMartin et al. (2018), MacMartin et al. (2022) and Parker & Irvine (2018). However, these avenues would violate the 1.5°C-target and, as suggested by Parker & Irvine (2018), depending on the amount of $Wm^{-2}$ compensated could take several decades. Therefore, even if SRM was, due to technical and / or environmental considerations, phased out earlier, the phase-out itself could become a multi-decadal undertaking in itself.

Being a reduced-complexity model, MAGICC7 has its caveats and constraints related to the physical and spatial resolution of
relevant climate and carbon cycle processes. Nevertheless, MAGICC has been used successfully in many instances to robustly analyse long-term perspectives (Meinshausen et al., 2020; Meinshausen, Smith, et al., 2011; Nauels et al., 2017). Therefore, we consider it also appropriate for this first quantification of hypothetical SRM deployment length which could motivate further study in more complex models.

It is important to highlight that even if global temperature was stabilised with the help of SRM, this will not provide a solution
with respect to regional impacts (A. C. Jones et al., 2018) and other impacts of high GHG concentration levels such as ocean acidification (Tjiputra et al., 2016). In our study, however, we do not aim to provide a comprehensive analysis of impacts or side-effects of such a climate intervention nor do we provide a likely or desirable implementation strategy but rather explore a concept to determine hypothetical SRM deployment lengths.

**5 Conclusions**

In this paper, we generate a large dataset of pathways to analyse SRM deployment length in scenarios that are consistent with 2030 NDC levels and use SRM to cool down to 1.5°C of warming. We find a large spread of SRM deployment lengths, ranging from no NNE and SRM requirements at all to massive NNE and SRM requirements past 2500 to limit global warming to 1.5°C. Most of our simulations require around 150-300 years of SRM. Deployment timeframes are considerably dependent on
mitigation, negative emissions and climate uncertainty yet none of these factors alone can predict its length: Large cumulative positive emissions lead to long SRM deployments. However, small cumulative emissions are no guarantee for short deployment timescales. Similarly, realisations that require large cumulative negative emissions lead to long SRM deployments. But, small cumulative negative emission requirements do not necessarily imply short SRM timeframes. A large part of this uncertainty can be attributed to the uncertainty surrounding eTCRE-up and eTCRE-down.
For all realisations that follow current NDC median 2100-warming projections (2.4°C), none deploy SRM for a shorter period than 100 years even under the most optimistic assumptions on the eTCRE-ratio. For SRM deployment to truly be 'temporary', carbon removal technologies need to be massively scaled up within a relatively short timeframe, except in cases of very low

emission requirements and extremely high negative ZEC. Larger average annual NNEs do generally imply shorter SRM timescales.

Our study shows that the range of possible deployment timescales is vast even if pathways start at a similar point at the beginning of SRM deployment because the evolution of mitigation under SRM, the availability of carbon removal technologies and the effects of climate reversibility are not precisely known. Since these effects will be mostly uncertain at the time of SRM initialisation, a precedent prediction of deployment length seems unlikely with possibilities ranging from decades to multiple centuries. This is a knowledge gap that must be considered before any SRM proposal is seriously considered.


**Code and data availability statement**

The code is publicly available at https://doi.org/10.5281/zenodo.7707967.

Raw data has been taken from the IPCC AR6 WGIII database (Riahi et al., 2022; Byers et al., 2022).

**Author Contribution**

All authors designed the experiments and SB and ZN carried them out. SB prepared the manuscript with contributions from all authors.


**Ethics Declaration**

The authors declare that they have no conflict of interest.

**Acknowledgments**

SB is supported by CERFACS through the project MIRAGE. AN, ZN, BMS and CFS acknowledge support by the European Union's Horizon 2020 Research and Innovation Programme; the respective Horizon 2020 projects are PROVIDE (Grant No. 101003687) for CFS and BMS, the project ESM2025 (Grant No. 101003536) for ZN and BMS and the project CONSTRAIN (Grant No. 820829) for AN and CFS.

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
