# Peer review of "The Deployment Length of Solar Radiation Modification: An Interplay of Mitigation, Net-negative Emissions and Climate Uncertainty"

_Earth System Dynamics, 2022_

## Author Comment (AC1)

**Authors' Responses**

**Reviewer Comments in black, authors' responses in red**

RC1:
The study investigates temperature overshoot in a novel "current commitments" scenario that achieves large-scale negative emissions (of 3 different levels) in the 22nd century and beyond. These scenarios would overshoot the 1.5C threshold for over 300 years. On top of these baseline scenarios the study implements SRM to keep temperatures to 1.5 C. They find, rather unsurprisingly, that in a scenario that exceeds 1.5C for centuries if SRM is deployed to keep temperatures below 1.5C then it would be deployed for centuries.

There is very little that this study would add to the literature. The study's core finding is obvious, and the specific numerical value arrived at is determined by the scenario assumptions made by the authors and the one model that is applied. Furthermore, beyond showing the scenario(s) that the authors have created, the study has only 2 results: the length of time that SRM is deployed and the time-evolution of the cumulative carbon flux due to SRM. In my judgment there is not a sufficient depth of analysis or novelty in this work for it to be publishable in its current form.

Beyond the limited depth of analysis and lack of novelty, the results of this study are determined by the scenario assumptions made by the authors and by the insufficiently described ensemble of MAGICC6 model variants. While the scenario is fairly well described and quite reasonable, it is only 1 scenario (with 3 different CDR endings). There would be much more to analyse and discuss if a wider range of more and less ambitious scenarios were presented. The ensemble of MAGICC6 variants determines the results but there is insufficient description of how this ensemble is generated, nor is the reader given a quantitative assessment of its ECS and carbon cycle characteristics relative to more complex models or expert judgments.

In my judgment there is not a sufficient depth of analysis or novelty in this work for it to be publishable in its current form.

We thank reviewer 1 (R1) for taking the time to comment on our study. The main issues pointed out by the reviewer are the insufficient depth of analysis, the lack of novelty and the limited number of scenarios.

In the revised paper we have addressed all these issues. We have greatly increased the number of scenarios. Instead of using one stylized scenario with 3 different magnitudes of CDR we are looking at all scenarios in the IPCC AR6 WG3 database

that are aligned with 2030-NDCs and have decreasing or stagnating CO2 emissions in the last 5 years of the 21st century. This amounts to 355 scenarios with a wide range of 2100 warming outcomes, climate policy and CDR assumptions. We run each scenario with all 600 ensemble members of MAGICC7, resulting in 213 000 scenario realisations.

MAGIC V7.5.3 comes with several major updates in atmospheric chemistry and carbon cycle components compared to MAGICC6 and is IPCC AR6 consistent. We included a more thorough analysis of the ensemble members and the range of climate sensitivity they cover. An additional co-author was brought on board with additional MAGICC and scenario development expertise.

Regarding the lack of novelty, we want to point out that to our knowledge this dataset is now the largest set of overshoot scenarios available for investigating SRM peak-shaving, and similar findings have not been presented elsewhere. Due to the more diverse range of scenarios and assumptions included, we were able to not only analyse potential deployment length but to also explore what this length is dependent on and how much of this is under human control.

We hope that with these major revisions we were able to address all of the reviewer's core concerns.

---

## Author Comment (AC2)

**Authors' Responses**

**Reviewer Comments in black, authors' responses in red**

RC2:

We thank reviewer 2 (R2) for taking the time to provide helpful comments on our manuscript.

The authors of this paper note in multiple places that little attention has been posed to the question of deployment timescales of SRM. Considering they cite other studies that have done similar analyses (and arrived to similar conclusions, but with far more deatil) one might wonder how many papers are enough before something has received more than "little attention". Nevertheless, I agree that many times the words "temporary measures" or "stopgap measures" may give the impression that we're talking about decadal deployments overall, whereas it is more likely that (were SRM ever implemented) the commitment would be longer than that, and definitely span more than one generation. And this has been discussed before.

While we appreciated the concern raised by R2, we disagree with the reviewer that the research question of this study has been discussed before in sufficient depth. While some papers provide SRM figures with a time axis and thereby touch on the length of SRM deployment in their setup, it has not been systematically explored before. Outside of the SRM research community it is not understood that SRM deployment could span more than just a few decades and there is no paper that discusses deployment timescales comprehensively.

However, when defining "how long" this "long" would be, one must rely on very long term assumptions about not just SRM, but climate policy and humanity in general. My fundamental issue with this paper is that the authors replicate partly what has been done in the extended RCPs and SSPs but without any of the carefulness employed by Meinhausen et al.: in the work around the extended scenarios, it is made clear multiple times that the goal is to look at the long-term Earth System response, and not to pretend to forecast how emissions (or CDR) will look like in 2250. It is a subtle distinction but an important one, especially if, as the authors do here, the simplistic assumptions around long term policies (i.e. "we extend the last 20 years of the century for 400 more") are used to come up with rather precise numbers over the timescales of SRM deployment where the uncertainty is only related to climate sensitivity. But the work of, for instance, Lehner et al. (2020) clearly indicate that by the end of the century the main source of uncertainties in CMIP6 projection is the one related to scenarios - and indeed this is why the IPCC spans multiple ones. Selecting one arbitrary scenario, pretending it is valid for 500

years (I appreciate the validity of scenario-building, even on the very long-term, but imagine extrapolating the last 2 decades of the sixteenth century to find out how humanity would have fared in the year 2000) and then pretending that can give us a hard, numeric idea of SRM timescales of deployment just because MAGICC has been used to derive it seems incredibly weak to me.

We want to thank the reviewer for pointing out the weakness of our scenario assumptions and agree that one stylised emissions scenario falls short of a robust representation of potential future SRM peak-shaving pathways and the large range of uncertainty related to socio-economic development. Therefore, we greatly reduced any underlying scenario assumptions in our revised manuscript. We are using all scenarios in the IPCC AR6 WG3 database that are aligned with 2030-NDCs and have decreasing or stagnating CO2 emissions in the last 5 years of the 21st century. This amounts to 355 scenarios with a wide range of 2100 warming outcomes, climate policy and CDR assumptions.

The reviewer also criticised the precision implied by the previous manuscript version and we regret that such an impression was made. We carefully revisited relevant statements to refrain from using any formulations in our revised manuscript that could be misunderstood.

The authors admit the same at the beginning of Section 4: different assumptions over GHG emissions and CDR would alter the outcome considerably. But what the authors dismiss as obvious, it is not. I don't see the merit of "determining" an outcome with a +/- 35 years precision that far into the future without including the context of other scenarios that span multiple sets of assumptions. Is a middle-of-the-road scenario that is currently tracking pledges the same scenario that would better track emissions at the end of the century? We can't know, and therefore the only way we have is exploring multiple future pathways - being extremely clear that they are idealizations most likely to be wrong. But what the authors do is try to pretend their assumptions are the only "reasonable" ones, and fail to highlight enough how arbitrary their results are.

We regret giving the impression that the selected scenario was the only reasonable one as this was not our intention and agree that we did not sufficiently highlight how sensitive the results are to other scenario assumptions. We hope that the inclusion of the wide range of scenarios as described above and the revised focus on many kinds of different hypothetical peak-shaving pathways will address R2s concern.

There are multiple parts of the manuscript where the authors make pretty arbitrary assumptions but try to pass them off as "neutral". For instance: "Specifically, we assume that the availability of SRM affects mitigation ambition and that after SRM

initialization there is no incentive to increase ambition beyond the currently pledged targets. It is of course impossible to know how emissions would evolve under SRM"

If it's impossible (and I agree) than how can one make that assumption so light-heartedly? Also, the authors say that unlike MacMartin et al. (2018) they consider mitigation, CDR and SRM in conjunction, but if you take an emission pathway that already exists and don't modify it based on the presence of other components (SRM), you are indeed considering them as independent additive components. I would suggest you look into Drake et al. (2021) for an example of an exercise trying to consider these aspects in conjunction. The authors can also discuss some recent results relating to economic games where the concept of SRM is introduced, which generally point towards opposite results from what the authors assume (there are two that just recently came out, Talbot et al., 2022; Todd et al., 2022 and more in the past). If the authors chose to ignore these results, they need to explain why.

Due to the different scenario setup of our new manuscript, we are not explicitly exploring any interaction between SRM and mitigation ambition. We thank the reviewer for pointing us towards interesting literature.

As another comment, in the abstract, the authors define their derived long times of deployment for SRM as a "risk". But a risk compared to what? If it's an abstract risk of just "doing" SRM (i.e. an almost ethical one: SRM is wrong and therefore the longer you do it, the more you are sinning) then it's the authors personal view. If it's a risk of negative effects (that the authors mention) and these risks trump those from climate change (which the authors don't mention, plenty of literature around comparing risk of deployment versus risks of not deploying) then one might imagine that SRM wouldn't hinder mitigation ambition but strengthen them.

Thank you, we are clearer in our revised manuscript on what kind of risk we are referring to.

In conclusion, I don't think this manuscript is suitable for publication on ESD in its current form. The authors analyze their own scenario, based on their own assumptions, come up with a number and then consider and explain that number as an inevitable conclusion of any kind of SRM deployment under any possible scenario. What novel insight does that shed on anything? The analyses related to the carbon cycle are also to my eyes unremarkable, especially when other analyses based on more comprehensive models are already available.

We hope that our substantial revisions will address the reviewer's concerns around the relevance of the study. We tried to be clear in the former manuscript version that the former results only hold under a current policy pathway with the selected assumptions. We therefore sincerely regret that the presentation of our previous results was misunderstood to generally hold for SRM deployment under all possible pathways. We are, however, convinced that the revised setup and result

presentation will not create this impression. Finally, we also agree that there are models far better suited for an analysis of carbon cycle effects under SRM and therefore removed this part in the manuscript.

I agree with Reviewer 1 that this work would be far more robust if multiple scenarios where analyzed: if stronger mitigation and more CDR was available, as in other IPCC scenarios, how would these results look like? If the stabilization target was 2 instead of 1.5? This could probably be a way in which this work could become suitable for publication, together with a much more in depth discussion of methods used and a broader overview of past literature on the subject.

Thank you for your comment, we have substantially revised the methods and now explore a very large set of scenarios as suggested. We have also produced a much more in depth analysis of the factors controlling SRM deployment length. We sincerely hope that our revised manuscript addresses your concerns by using this revised, much more comprehensive and nuanced study design .

---

## Author Comment (AC3)

**Authors' Responses**

**Reviewer Comments in black, authors' responses in red**

RC3:
**General Comments:**

They key point of the article is an important one: under a realistically pessimistic continuation of the new "business as usual" and feasible CDR rates, keeping warming below 1.5°C would require centuries of SRM before CDR could clean up the carbon mess. While I agree with **RC1** that this point is obvious to experts, this does not mean that the obvious should go unstated! **(RC2** somewhat agrees but finished by saying "this has been discussed before", without citing any specific examples.) I find the discussion in the literature unsatisfying and, at worst, deliberating misleading in a way that portrays SRM as less problematic than it really is.

We are thankful to hear that R3 thinks that our study represents a valuable contribution to the SRM discussion and fully agree with the state of published literature on this issue.

However, I did find much the paper to be much longer than it needed to be; many paragraphs are unnecessary distractions from the key point. I also agree with **RC1** and **RC2** that the authors are trying to pass the scenarios as much too sophisticated or realistic. In contrast to **RC1** and **RC2,** however, I would instead encourage the authors to dramatically simply their scenario and methods and submit it as a short "perspective", either in Earth System Dynamics or elsewhere. For example, they could use idealized emissions scenarios that are constant positive emissions —> linear decrease with prescribed slope —> constant negative emissions; these scenarios would be characterized by just 3 constants (the year emissions reductions begin; the emissions reductions rate; the CDR level) and the authors could briefly explore how these parameters affect the SRM timescale and how they relate to various scenarios discussed in the literature (including the somewhat realistically-pessimistic scenario they described here). While I do not think this article merits publication in Earth System Dynamics as is, I think that a revised article that is entirely (and only) about SRM timescale could merit publication.

We want to thank the reviewer for the suggestion to submit the study as a "perspective". We eventually decided to go down a different route by greatly expanding the analysis to include many more scenarios to fully counter the impression that results from our single scenario design could be generalised. More

specifically, we increased the number of scenarios to 355, spanning a wide range of 2100 warming outcomes, climate policy and CDR assumptions.

We hope that with these substantial revisions and expanded methodological setup, you will agree that the article is suitable for a publication in ESD.

**Specific Comments:**

I do not think the carbon cycle discussion merits inclusion in the article. The feedback is tiny compared to cumulative emissions and CDR, probably very model specific, and is not at all explained in terms of underlying processes.

We removed the carbon cycle discussion in the revised manuscript.

**RC2** mentioned that Drake et al. 2021 model scenarios with tradeoffs between mitigation, CDR, and SRM. Belaia et al. 2021 do as well (https://www.worldscientific.com/doi/full/10.1142/S2010007821500081). Neither *explicitly* state a timescale over which CDR has to be maintained, but it is obviously multiple centuries from their plots.

Given the discussion of climate changes on long timescale, it may be worth mentioning idealized scenario modelling that includes Sea Level Rise, e.g. Montero et al. (https://egusphere.copernicus.org/preprints/2022/egusphere-2022-135/).

Line 102: It should be explained where the 2.2 Wm^-2 value comes from.

Line 109: Is the 0.12 Wm^-2 relative to some long-time mean? What does this number signify?

Thank you for these specific comments, we have taken them into consideration for our new manuscript.

---

## Author Response (AR2)

**Authors' Responses**

**Reviewer Comments in black,** authors' responses in red

Let me start by saying that I acknowledge this review process must have been particularly hard to the authors due to the three long and in some ways rather conflicting reviews. I really admire the depth the authors have gone to revise the paper, consider the feedbacks, and produce this new version. I am particularly happy the authors decided to follow the advice and consider a wide range of scenarios now, making their analyses much more relevant and useful.
This is indeed a completely different paper from the first version, and a far, far better one at that. I recommend acceptance for publication in ESD and only have a very few comments below. I think this is now going to be a great addition to the literature! Also, great title!

We are grateful for Mr. Visionis review of our resubmitted manuscript and very happy to have successfully addressed the referees concerns with our revisions.

Some comments:

L 23: "pan-generational" doesn't sound like a very scientific term

We have changed it to "multi-generational".

L 24-25: this is just a repetition of what you just said in the phrase before. I would use the abstract to discuss the results a bit more in depth (as you do in the conclusions, there are plenty of other facts from your study that can go here!) and not make twice the same point.

We want to thank Mr. Visioni for pointing out the repetition and have updated the last part of the abstract including removing the second to last sentence. The last sentences now read: "Since the evolution of mitigation under SRM, the availability of carbon removal technologies and the effects of climate reversibility will be mostly unknown at its initialisation time, it is impossible to predict how 'temporary' SRM deployment would be. Any deployment of SRM therefore comes with the risk of multi-century legacies of deployment, implying multi-generational commitments of costs, risks and negative side effects of SRM and NNEs combined."

The introduction now does a much better job at framing the problem and presents a much more balanced overview of the issue – thanks for updating the references. Also, they clearly now state what the study does and acknowledge the limitations.

The way in which now you extrapolate to 2500 makes much more sense and is really robust, whatever one might think of extending these scenarios so far away in

the future. Figure 1 does a nice job at explaining what you've done, but can be improved: enlarge the single panels a bit, separate them clearly with borders, and perhaps add arrows (which would entail a reordering) to more clearly indicate the "extension" process for a-b-c and d-e-f and how they then combine in panel g.

We are very happy that the introduction now fulfills the referees expectations and that the revised scenario extensions are appreciated. We have updated the figures as suggested.

Line 171 – replace "earliest" with "no sooner than"

Done

Figure 2, same for Figure 1. It is better to "frame" the different plots in different boxes otherwise it might be hard to follow. But otherwise this figure is so clear and nice!

Done

Line 315 - deployments

Done

Line 331 – reference needs to be in brackets

Done

Line 368-369 - I would say this ethical risk needs to be balanced out with the idea of imposing climate change-related risks, forced migrations and perhaps unbearable conditions to generations who did not contribute to the problem to begin with.

We want to thank Mr. Visioni for raising this issue. We have added: "This ethical risk needs to be considered in conjunction with the additional climate change-related risks from ongoing warming also imposed on future generations who did not contribute to the problem to begin with."

Line 376 – no comma after "both"

Done

Line 380 – also see MacMartin et al. (2022), where a phase-out scenario is explicitly simulated.

MacMartin, D. G., Visioni, D., Kravitz, B., Richter, J., Felgenhauer, T., Lee, W. R., Morrow, D. R., Parson, E. A., and Sugiyama, M.: Scenarios for modeling solar radiation modification, P. Natl. Acad. Sci. USA, 119, e2202230119, https://doi.org/10.1073/pnas.2202230119, 2022

Thank you for pointing us to this specific literature reference, we have added it.